# Global Food Loss and Waste in Primary Production: A Reassessment of Its Scale and Significance

**Julian Parfitt \*, Tim Croker and Anna Brockhaus**

Anthesis Group, Unit J, Taper Studios, 175 Long Lane, Bermondsey, London SE1 4GT, UK;
Tim.Croker@anthesisgroup.com (T.C.); anna.brockhaus@anthesisgroup.com (A.B.)
**\*** Correspondence: julian.parfitt@anthesisgroup.com

**Abstract:** Global statistics on food waste were first reported by the United Nations Food and Agriculture Organization in 2011, and since that time, more attention has been given to food waste measurements at the consumer, retail and hospitality stages, whilst efforts to quantify losses during primary production have been more limited. To provide an updated view of global losses in primary production, data for the harvest and on-farm, post-harvest stages were reassessed through a systematic review of data sources and a selection of datasets for further analysis. To qualify for selection, food-loss measurements needed to be specific to primary production and to particular food commodities and production regions. The analysis covered a split between losses at the harvest and post-harvest stages linked to activity descriptions within the primary data sources. A cross-sectional sample of ten commodity/region case studies was conducted through stakeholder interviews and literature reviews to triangulate food waste estimates and to understand issues relating to food waste definitions from a farming perspective.

**Keywords:** food waste; food loss; harvest losses; post-harvest losses; primary production; SDG 12.3





## 1. Introduction

Food waste reduction is an important component of the more sustainable agrifood systems that are needed to address global environmental challenges [1] and to deliver across a range of United Nations sustainable development goals, including zero hunger, poverty, food waste reduction and biodiversity [2]. The primary-production stage has been identified as a major hotspot for global food waste but has not received the same attention as other key points in the supply chain [3]. In 2011, the FAO published the first global assessment of food waste with all supply-chain stages in scope, including harvest and post-harvest losses during primary production [4]. A decade later, this study remains the only reference for global losses in primary production that includes both harvest and post-harvest food waste and is the source of the often-quoted statistic that approximately a third of the edible parts of food produced for human consumption is lost or wasted globally, equivalent to 1.3 billion tonnes per year.

Reasons for the limited knowledge of primary-production food losses include the paucity of data from in-field measurements [5] and the difficulty in the application of standard food-loss definitions to the varied circumstances of primary-production systems [6,7]. Furthermore, the lack of focus on the farm stage in more affluent regions has been reinforced by the view established by the 2011 FAO study that consumer food waste is the predominant source of food waste. Conversely, losses during the primary-production to market stages were reported to be more significant in lower-income regions [4]. For consumer food waste, this assumption has recently been challenged by the findings of the Food Waste Index Report, where higher rates of consumer food waste were identified in less affluent countries [8]. In the EU's approach to food waste reporting by member states, post-harvest losses at the farm stage are included in reporting requirements, but harvest

losses are included on a voluntary basis [9]. This is consistent with the UN FAO's Food Loss Index, which excludes harvest losses in tracking progress towards SDG 12.3 [10].

Against this backdrop, the current study was commissioned to reassess the scale and impact of global farm-stage food losses, including harvest and post-harvest elements, as illustrated in Figure 1. This approach was designed to make the most effective use of data collected since the FAO 2011 study whilst also highlighting where data gaps limit the reliability of estimates and hotspot identification.

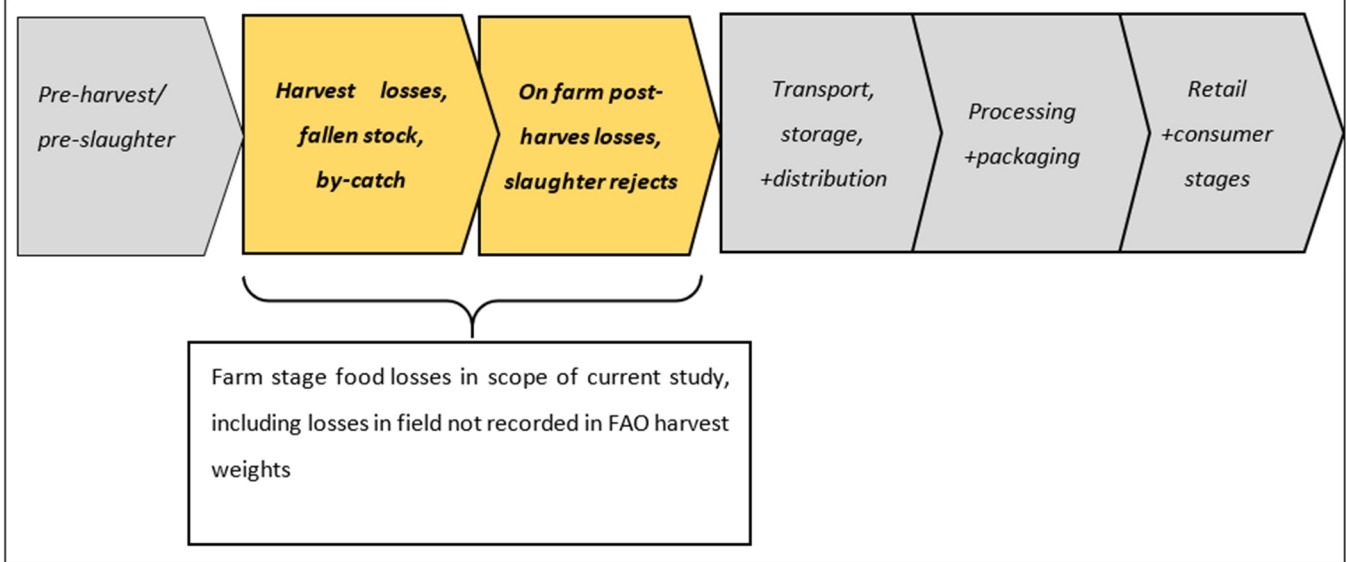

**Figure 1.** Study scope.

Throughout the text, the terms 'food loss' and 'food waste' are used interchangeably and are not intended to designate differences in food waste drivers, as posited by some authors and adopted by the FAO's Food Loss Index [10]. Such a distinction has been used to reinforce links between 'food waste' resulting from more conscious decisions to discard at the retail and consumer stages and 'food loss' resulting from inadvertent food waste linked to poor infrastructure, pests, disease and other factors over which actors towards the beginning of the supply chain have little control [11]. The authors believe that such a distinction distorts the wider understanding of how food waste drivers are linked across supply-chain stages.

## 2. Methodology

The work identifies the scale and global profile of farm-stage food losses using a four-stage methodology (Figure 2) to combine selected datasets to produce high-level estimates of food waste across commodity groups and global regions. For ten global commodity-region pairings, case studies involving stakeholder interviews and literature reviews were conducted to understand the limitations of the assumptions made in the quantification process and to provide insights into waste drivers.

### 2.1. Defining Primary Production

Data sources were selected according to a set of identifiable harvest and post-harvest activities described within the scope of each data source identified for the study. Table 1 illustrates the range of activity descriptors associated with harvest and post-harvest losses within scope (Figure 1), with many post-harvest activities overlapping with subsequent supply-chain stages beyond the farmgate. Determining these boundaries is, in part, related to the context of each study and to crop/commodity types and their perishability. Overall, this followed a gradient between smallholder farmers in poorer economies, with much of

the processing carried out on-farm, versus larger-scale, more highly industrialised systems, where more processing activities are carried out post-farmgate.

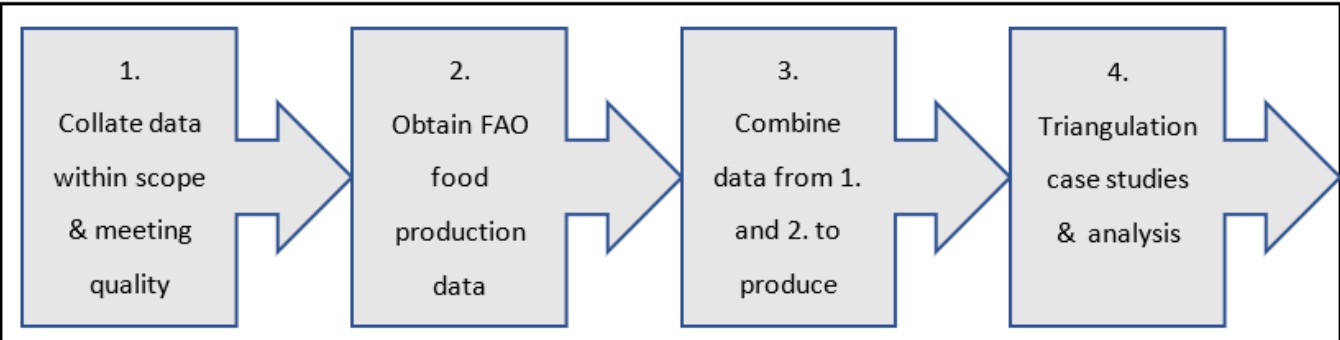

**Figure 2.** Steps in food-loss quantification methodology.

**Table 1.** Activities associated with food losses during primary production extracted from reviewed data sources.

| | | |
|---|---|---|
| Cleaning | Harvesting | Removing cobs from stalks |
| Collection from field | Lifting (e.g., cassava) | Shelling (e.g., groundnuts) |
| De-husking | Milling on-farm | Sorting/Grading |
| Drying (e.g., rice, fish) | On-farm assembling | Stacking |
| Drying Before Storage | On-farm storage | Stooking |
| Drying Before Threshing | Packing on-farm | Storage |
| Farm storage | Piling | Storage, handling |
| Field drying | Platform drying | Stripping |
| Fishing | Pulping | Threshing |
| Grading | Preliminary Processing | Transport on farm |
| Grading & Sorting | Processing on-farm | Winnowing |
| Handling | Rearing of livestock | |

In addition to the activity-related classification, harvest losses identified in the literature and databases also included crops left unharvested, relating to terms such as 'plough-back', 'left in field' and 'walk-by'.

Although the objective was to classify data into harvest and post-harvest activities, overlaps and uncertainties exist around this boundary, depending on how food losses were reported within each study. For example, preliminary in-field grading of vegetables may be described as a crop 'left in the field' in one study (a harvest loss) or separated out in another as a post-harvest activity.

## 2.2. Application of Food Waste Definitions to Primary Production

Consistency in the application of food waste definitions a prerequisite for successful benchmarking and tracking of progress against the SDG 12.3 food waste reduction target. Food waste is defined by the Food Loss and Waste Accounting and Reporting Protocol (FLW Protocol) [12] as any foods that are, or were at some point, intended for human consumption but which are either not harvested or sent to one of eight food waste destinations (Figure 3). Food waste may exclude inedible parts but may not include losses that occur prior to harvest as currently, there is no agreed methodology relating to these losses within the Protocol.

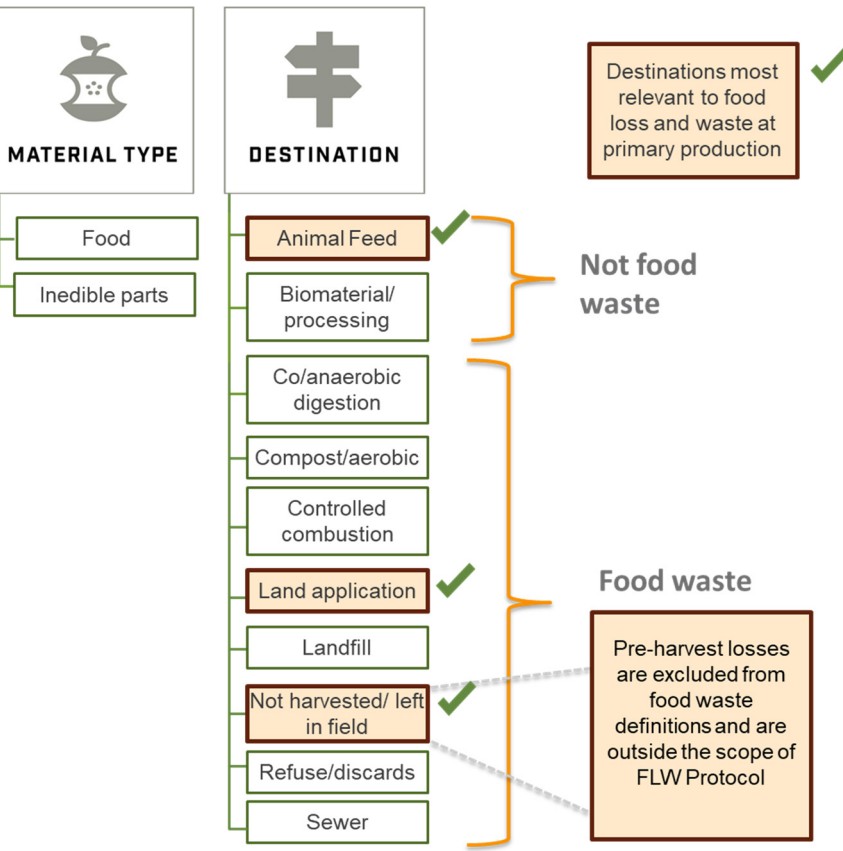

**Figure 3.** Food waste definitions under FLW Protocol indicating destinations most relevant to primary production (adapted from [12]).

Food waste is defined from the point at which outputs from primary production can be regarded as 'food' based on the intended use. It does not include losses from animal or plant products produced for non-food purposes, such as industrial uses, feed and seed production. For crops and produce, this is defined in terms of crop maturity and being 'ready for harvest' [7]. For livestock and fisheries, a similar definitional principle is applied by the FLW Protocol (i.e., based on maturity, slaughter weight or when wild caught animals/fish are harvested) but is more problematic to interpret in practice. In reviewing datasets for this research, it was noted that fallen stock or poultry 'dead on arrival' at slaughter may be recorded by studies, but the question of animal 'maturity' and whether the loss would count as 'food' remains largely theoretical. Farm-stage research into food losses is rarely designed with definitional frameworks in mind, and most of the available studies identified by the research were conducted before FLW Protocol definitional rules were established in 2016.

Food waste definitions may differentiate the material type being measured as edible and inedible fractions of food (Figure 3), with the inedible fraction, by its nature, not contributing to the nutritional value as food. Within this research, the inedible components were excluded from food waste totals based on the assumptions used to support estimates made by the 2011 FAO study [13]. In common with reporting practices for meat production, livestock weights were expressed as carcass or 'dressed' weights: the weight measured after being partially butchered (i.e., removal of internal organs, head and other inedible portions of the tail and legs).

### 2.3. Collation and Selection of Data

2.3.1. Selection Criteria

Criteria were developed to assess the suitability of available food waste data for inclusion in the study. These covered data scope, timeliness, commodity and region specificity, and significance to primary production within region.

Scope of data:

- Data source contains food waste estimates that relate to pre-farmgate, including % loss at harvest and during post-harvest stages.
- The focus was on quantitative loss of food by weight, as too few studies contained measurement of loss in food quality or value.
- Those data points covering both harvest and post-harvest losses needed to differentiate losses by farm stage rather than providing a combined statistic.
- For livestock, losses were included prior to slaughter.
- Whole-chain studies that did not contain separate estimates of losses at the farm stage were excluded.

Timeliness:

- In order to provide an update on the 2011 FAO study and to be relevant to current primary production practices, data were selected for their timeliness.
- Except for a few commodity regions poorly represented in the data, 92% of selected data points were more recent than those compiled for the FAO 2011 study (2008 or later) [13].

Commodity and region specificity:

- Sources were selected that contained estimates of losses for specified commodities and geographies.
- Food waste data relating to generic groupings, such as 'fruit and vegetables', were excluded.

Significance to primary production within region:

- Datasets were assessed for their significance to agricultural-system and commodity types within each region. Representation of the predominant commodity types for a particular region was important in scaling estimates of loss with FAO country-level production data. Particular attention was paid to differentiating data derived from more manually based agricultural systems from more highly mechanized systems in regions where both system types were collocated.

2.3.2. Data Sources

Farm-stage loss studies were compiled from existing databases and literature searches [14–16] for data that met the selection criteria. The main global data source for food waste is the FAO's open-access online database, containing 18,000 observations, including entries for losses during primary production [15]. This source was downloaded, deduplicated and coded for the purposes of this study. A literature search was conducted to identify additional data sources, including searches within World Resources Institute/WRAP's Food Waste Atlas [16], academic literature and online reports, including 'grey literature', governmental/NGOs publications, trade bodies, PhD/MSc theses, conference papers and book chapters. Use was also made of publicly reported food waste data from growers under the Champions 12.3 $10 \times 20 \times 30$ initiative [17].

The collated records comprised 20,000 datapoints, of which 3816 met the selection criteria and 2172 provided usable harvest or post-harvest loss factors. Table 2 profiles the selected datasets by commodity group and by region. Commodity groups and global regions are defined in Appendices A and B.

**Table 2.** Number of farm stage studies meeting selection criteria, by FAO 2019 commodity group and region.

| | Cereals & Pulses | Fruit & Vegetables | Meat, Poultry & Dairy | Roots, Tubers & Oil Crops | Fish & Seafood | Other | Total |
|---|---|---|---|---|---|---|---|
| Europe | 5 | 66 | 60 | 42 | 2 | | 175 |
| North America & Australasia | 2 | 27 | 5 | 7 | 10 | | 51 |
| Industrialised Asia | 15 | 38 | 1 | 3 | | | 58 |
| Sub-Saharan Africa | 1060 | 103 | 5 | 34 | | | 1202 |
| Northern Africa & Middle East | 6 | 8 | 7 | 11 | 1 | | 33 |
| South & Southeast Asia | 165 | 231 | 21 | 117 | 11 | 57 | 602 |
| Latin America | 23 | 9 | 3 | 9 | 5 | | 51 |
| Total | 1276 | 4822 | 104 | 224 | 29 | 57 | 2172 |

In the matrix of 42 cells shown in Table 2, 8 commodity-region pairings had no available data. These data gaps related to fish and seafood and the miscellaneous group that included many crops not grown in temperate regions (see Appendix A). Although gaps in the data present a challenge to the scaling of food waste quantities for each commodity-region, the available data is an improvement on the FAO's 2011 study, in which 46% of cells relating to primary production within the commodity-region matrix had no datapoints [13].

Two global regions accounted for the majority of datapoints, with 55% relating to Sub-Saharan Africa and 28% to South and Southeast Asia. Cereals and pulses accounted for 59% of all datapoints, (83% of these related to Sub-Saharan Africa) and fruit/vegetables accounted for 22% (48% from South and Southeast Asia).

A range of different primary-data collection methods were apparent in the selected data sources (Figure 4). Most involved mixed methods of measurement (e.g., 57% used both in-country expert opinion and an element of sample surveys), with heavy reliance on surveys combining interviews and questionnaires. Due to the expense of conducting on-farm measurements, only 3% of datapoints were based solely on this technique. Methodology has important implications for the consistency and reliability of loss estimates, with self-declared losses generally under-reporting food waste compared with data obtained from direct measurement in the field [5].

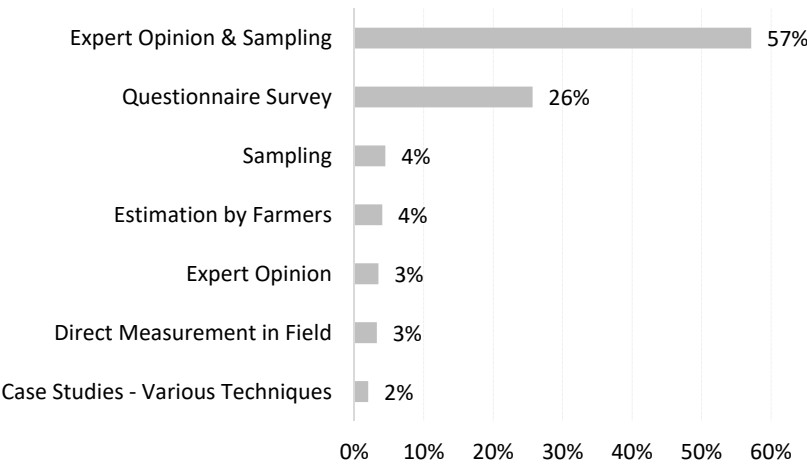

**Figure 4.** Methodologies used by farm-stage food-loss studies meeting the selection criteria.

### 2.4. Integration of Food Loss Data with Food-Production Statistics

The final dataset was used to compute a directory of percentage losses by commodity and country, with values allocated to primary-production stages on the basis of activity descriptions within each study (e.g., animal losses as fallen stock pre-slaughter, harvest losses and post-harvest losses).

The production data across all food commodities were collated from FAOSTAT data sources (Table 3) for the year 2016 (the most recent year with complete data), covering primary production for all countries and regions in the world. These data were then mapped onto the commodity lists used to compile the food waste data (Appendix A, Tables A1 and A2).

**Table 3.** Food production statistics: sources used for scaling farm-stage food waste.

| Source | Link |
|---|---|
| FAOSTAT Livestock—Primary Production Volume Statistics | http://www.fao.org/faostat/en/#data/QL (accessed on 15 July 2021) |
| Crop Production Volume Statistics | http://www.fao.org/faostat/en/#data/QC (accessed on 15 July 2021) |
| FAOSTAT FBS Fish and Fish Products Volume Statistics | http://www.fao.org/faostat/en/#data/FBS (accessed on 15 July 2021) |

FAOSTAT production data represent the weight of all crops or livestock after harvest or slaughter and cannot be equated to the quantity of edible food available after harvest. In order to convert the production data into edible food, two different factors were applied:

1. Human food supply allocation factor to determine the part of agricultural production that is allocated for human consumption. These factors were derived from the technical report relating to the 2011 FAO study [13], with each region allocated a different factor, which were applied to estimate the total production kept within the human-food supply chain (i.e., excluding quantities sent to animal feed or other non-food uses, such as bioethanol production).

2. Edible food conversion factor applied to total production for human consumption to estimate the edible-food fraction. Of the total quantities for human consumption, the edible fraction was estimated based on fractions assumed to be edible and allocation factors for the part of agricultural production that is allocated to human consumption compiled for the 2011 FAO study [13].

The percentage-loss averages were coded to detailed commodity types (Appendix A, Table A2) and global region (Appendix B). These were then matched to FAO production data with the objective of finding the most accurate match possible (i.e., first mapping at the commodity type within that region, and otherwise mapping to the commodity-group average within that region). Those commodities with large production tonnages within a particular region were given extra scrutiny within the review process. Where there were data gaps or insufficient representation for a commodity group, appropriate substitute values were used, based on data obtained from comparable regions or from the closest commodity group.

To calculate the food waste tonnage, inclusive of losses in the field or around harvest, an adjustment was made to the FAO production statistics to take account of agricultural production lost before FAO production (harvested) weights were obtained. Thus, 'harvested weight' from FAO data represents a lower tonnage than 'total agricultural production'.

### 2.5. Triangulation Case Studies

Ten case studies were carried out to sense check food-loss calculations for selected commodities and regions and to explore the underlying complexities of losses associated with primary production. These were selected to provide a broad cross-section of different commodity-region pairings and to reflect differences in data availability and agricultural

systems covered (Appendix C). The study carried out 20 interviews across different stakeholder groups, including with NGOs, trade associations, primary producers and research institutes. Of these, 13 interviews were specific to commodity regions, and 7 explored overarching themes, such as field measurement, whole-chain initiatives and food waste drivers of farm-stage losses. Expertise relating to farm-stage losses is fragmented and not easily accessed, so it was not possible to complete interviews for all case studies. Further evidence gathering involved an extensive literature review that located over 60 relevant publications. Of greatest value was case-study literature that placed an emphasis on direct field studies aligned to the FAO's 'four elements approach' to food loss analysis:

(1)   screening (for known research literature and consultation with experts to gain an idea of the range of waste and main causes);
(2)   survey (including observational, group interviews, stakeholder interviews);
(3)   sampling (load tracking, field measurement, analysis of loss by activity); and
(4)   synthesis (involving root-cause analysis and solution identification) [18].

## 3. Results

### 3.1. Scale of Global Farm-Stage Losses by Weight

Global farm-stage food losses, including harvest losses and post-harvest losses, were estimated to be 1.2 billion tonnes per year. This was equivalent to 15.3% of total agricultural production intended for human consumption (including harvest losses). Table 4 shows that similar quantities were lost across harvest and farm-stage post-harvest activities, with 8.3% of total agricultural production lost in-field or around harvest (638 million tonnes) and 7.0% (537 million tonnes) during farm-stage post-harvest processes.

**Table 4.** Comparison of global food-loss estimates: current study compared with FAO 2019 Food Loss Index estimates [2]; food loss as % of total agricultural production/as % total harvested weight.

| | Current Study: Food Loss as % of Total Agricultural Production Including Field Losses and Harvested Weight | Current Study: Food Loss as % of Harvested Weight | FAO 2019 Food Loss Index Farm to Retail [2] |
|---|---|---|---|
| Harvest losses | 8.3% | 9.0% | Not included in FAO 2019 assessment |
| Post-harvest losses | 7.0% <br><br> Farm stage | 7.6% <br><br> Farm stage | 13.8% <br> Farm stage and in supply chain, up to but excluding retail |
| Total | 15.3% | 16.6% | 13.8% |
| | Excludes post-farmgate losses | | Excludes farm-stage harvest losses |

In 2019, the FAO published an assessment of global food losses [2] and estimated that 14% of global food production is lost across all post-harvest stages, from farm up to but not including the retail stage. The 2019 FAO report did not include harvest losses within its scope, so estimates are based on the weight of harvested crops rather than total agricultural production that included field losses. Additionally, the estimate is not directly comparable to the current study's 7.0% farm-stage post-harvest loss estimate, as it included post-harvest losses beyond the farmgate (Table 4). With the estimates from the current study as a proportion of total harvested weight re-baselined, the loss rate from on-farm, post-harvest activities, would be 7.6% and the equivalent of 16.6% losses at the farm stage, if harvest losses are also included on the same basis. Although it is not possible to combine these different estimates with the additional post-farmgate elements included within the 2019 FAO study [2] due to differences in methodology, the data suggest that between 20 and 25% of global production may be lost across the primary-production and supply-chain stages, up to but not including retail. Given the prevalence of self-reporting rather than

direct measurement within underlying farm-stage studies (Figure 2), actual loss rates are likely to be higher due to the tendency of questionnaires and indirect measurement techniques to underestimate harvest and farm-stage post-harvest losses [5,19].

The estimates for global losses during primary production from the current study generally support the findings of the World Resources Institute's 2019 assessment that identified food production and harvest as global food-loss hotspots [3], with on-farm losses significant across a range of settings, including within the agricultural systems of higher- and middle-income countries.

More detail of the scale of farm-stage losses is provided in terms of total tonnes (Figure 5), as a proportion of agricultural production (Figures 6 and 7) and as per capita losses (Figure 8). Figure 5 also indicates the uneven availability of food-loss data points across global regions, with two regions, Sub-Saharan Africa and South and Southeast Asia, accounting for 83% of usable data included in the review.

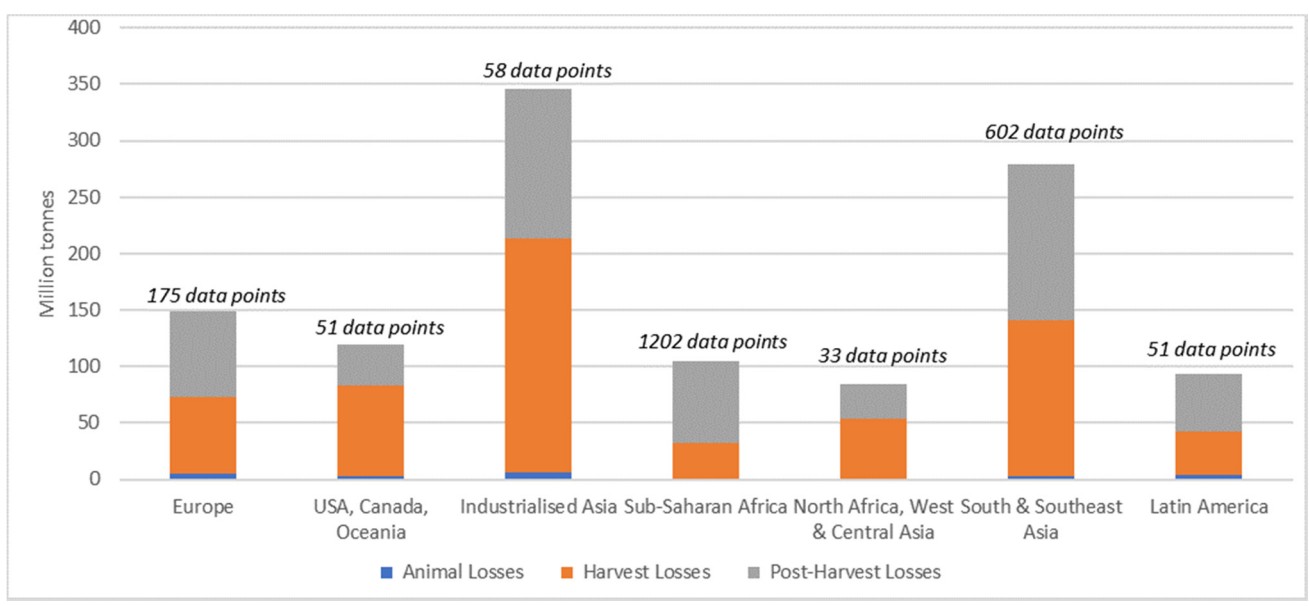

**Figure 5.** Food losses by stage and region (million tonnes) and indication of number of available data points.

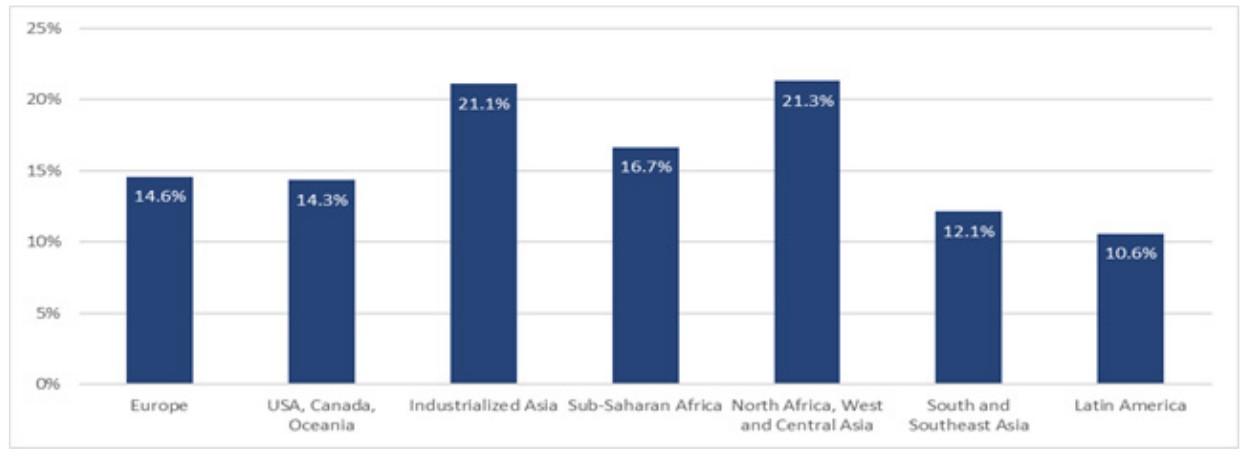

**Figure 6.** Farm-stage food losses by region as % total food production intended for human consumption.

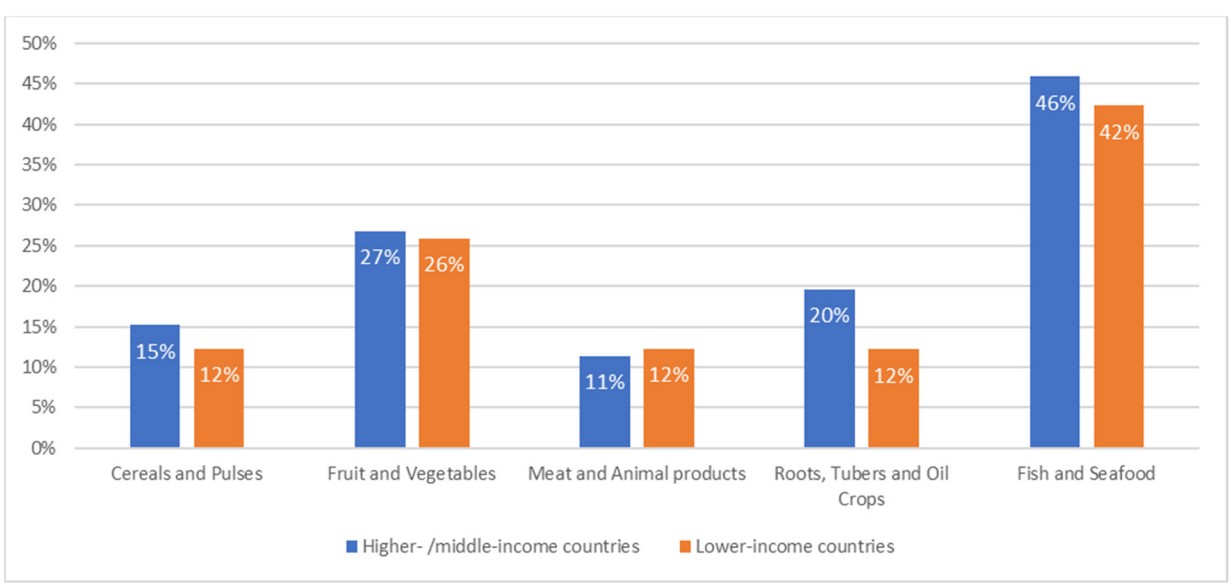

**Figure 7.** Farm-stage food losses by commodity group as % total food production intended for human consumption, by higher-/mid-income versus lower-income regions.

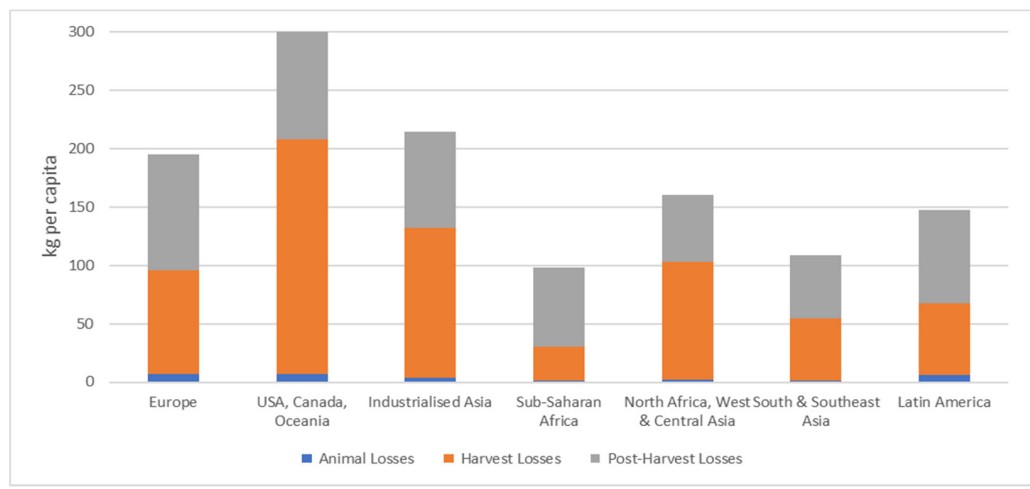

**Figure 8.** Per capita farm-stage food losses by region (kg per capita/year).

Figure 7 provides a summary of the proportion of commodity production lost across higher- and middle-income regions compared with lower-income regions. Except for meat and animal products, the proportion lost is higher within more affluent regions. Higher- and middle-income countries of Europe, North America and industrialised Asia, with 37% of the global population, contribute 58% of global harvest losses (368 million tonnes). Conversely, low-income regions, with 63% of the population, have a 54% share of global post-harvest farm-stage losses (291 million tonnes). The latter mainly relates to losses arising in Sub-Saharan Africa and South and Southeast Asia. The cutoff between post-harvest operations at the farm stage will differ across global regions, with more processing and sorting operations occurring pre-farmgate in lower-income countries, with a higher proportion of post-harvest losses in more affluent countries occurring in post-farmgate processing operations.

It is also important to note that in higher- and middle-income countries, the production of more perishable commodities per capita (fruit, vegetables, meat, dairy, fish) is approximately twice that of low-income regions, based on analysis of FAOSTAT production data. This is one factor amongst many explaining the variation in per capita farm-stage losses by region (Figure 8), which are generally higher in higher- and middle-income

regions (200–300 kg per capita/year) compared with lower-income regions (100–150 kg per capita/year).

Figures 9 and 10 display the same analysis split by commodity group. Key results are the significant scale of 'fruit and vegetable' losses in Industrialised Asia (tonnes, kg per capita and value) and the high per capita losses across multiple commodity groups for the United States, Canada and Oceania region.

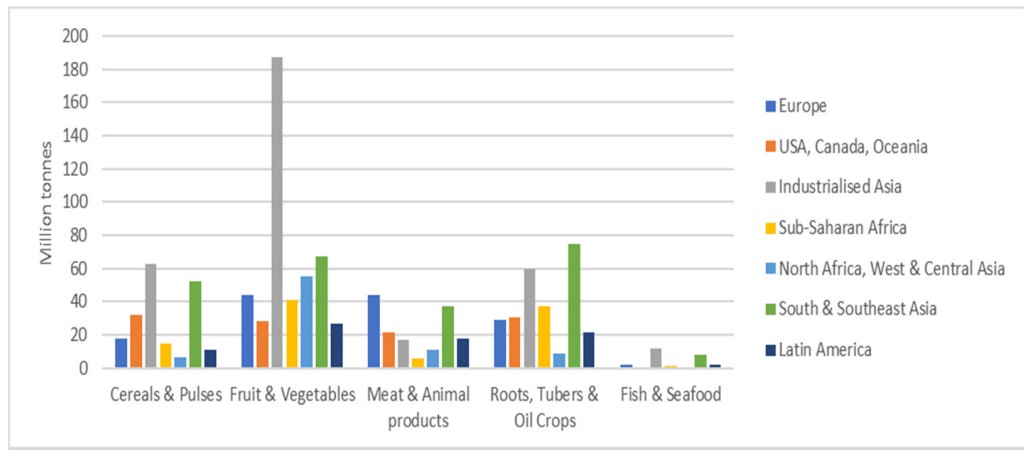

**Figure 9.** Food losses by commodity and region (million tonnes).

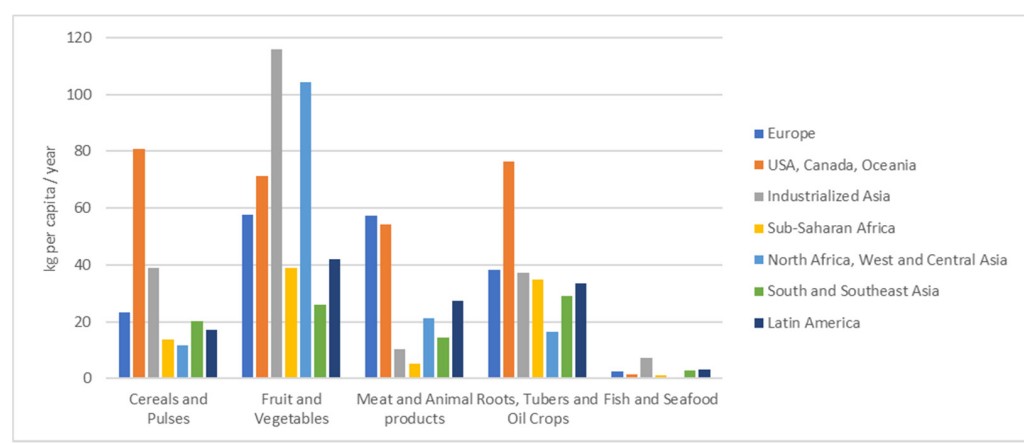

**Figure 10.** Per capita farm-stage food losses by commodity and region.

### 3.2. Scale of Global Farm-Stage Losses by Economic Value

The total value of global food losses, based on output prices at farmgate (FAOSTAT Value of Agricultural Production data), is estimated to be $370 billion, expressed as international dollars [20]. This is within the same ballpark as the $930 billion reported for whole-supply-chain food losses by the FAO in 2014 [21], considering that the previous estimate included the higher added value of losses in the supply chain beyond the farmgate (e.g., retail and consumer stages). Both sets of estimates exclude those from fisheries, as these are not covered by FAOSTAT data.

Figures 11 and 12 show the lost economic value of food loss in terms of farmgate prices by commodity group and region in absolute value and expressed as per capita value. Of note is the high per capita value of 'meat and meat product' losses in Europe, North America/Oceania and Latin America and the high per capita losses of fruit and vegetables associated with higher- and medium-income regions, as well as North Africa, West and Central Asia.

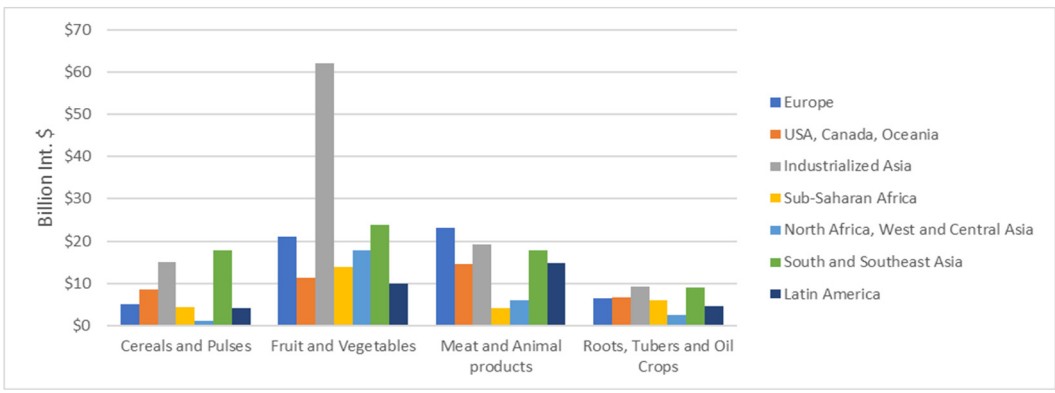

**Figure 11.** Food losses—value by commodity and region (billion international dollars, Int. $), excluding fish and seafood.

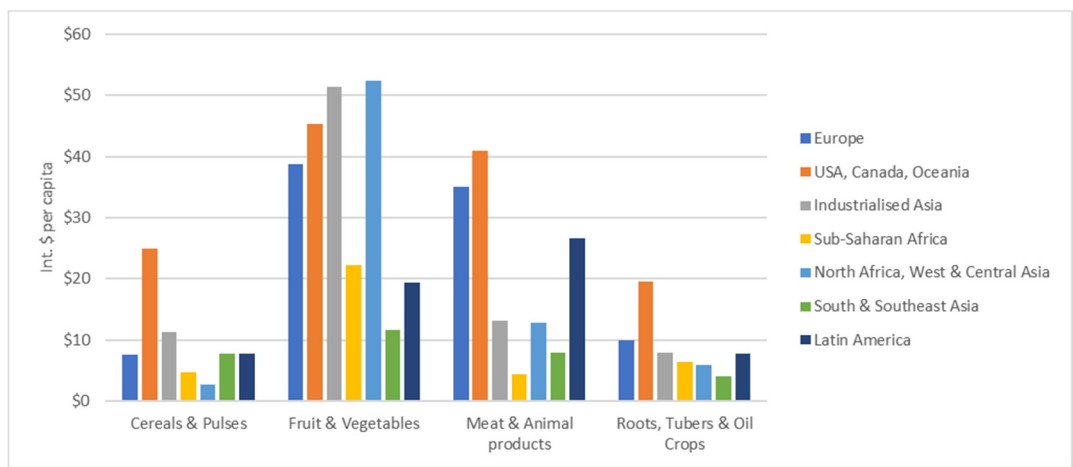

**Figure 12.** Food losses—value per capita by commodity and region (Int. $ per capita/year), excluding fisheries.

### 3.3. Triangulation of Global Food-Loss Results through 10 Case Studies

The 10 case studies contained 12 distinct supply chains (case studies 3 and 4 having two each), providing an assessment of farm-stage losses for their respective commodities, as summarised in Table 5. The interviews and literature reviews collated through the case studies were crosschecked against the relevant mean rates of food loss applied within in the global food-loss calculations. The assessment also considered the relative availability of datasets, ranging from 'very poor' (with few/no data points available for the case-study commodity within the region) to 'very good' (a large number of data points). Only two regions had cases that scored 'very good' for any of the relevant commodities: South and Southeast Asia and Sub-Saharan Africa, covering rice, fruit/vegetables and groundnuts. Scores did not take account of the farm-level methodologies used to collect primary data as it was established in the review of datasets that most measurement involved self-reporting and expert opinion rather than more robust methods involving an element of in-field measurement (Figure 2).

Table 5. Summary of results from triangulation case studies ('PHL' = post-harvest loss).

| | Case Study Commodity Region | Commodities Covered | Relative Availability of Datasets Identified from Data Review Stage of Current Study | Food Loss % Mean Value Derived from Available Data | Case Study Evidence Supports Lower Loss Rates | About Right | Case Study Evidence Supports Higher Loss Rates |
|---|---|---|---|---|---|---|---|
| 1 | Europe—cereals and pulses | UK—wheat | very poor | 10.00% | 1.30% | | Inclusion of grain to animal feed within source studies? |
| 2 | S and SE Asia—cereals and pulses | Pakistan and India—rice | very good | 13.10% | | 10.00% | |
| 3 | Sub-Saharan Africa—citrus and vegetables | S-S Africa small-holder farms—citrus and veg. | very good | 48.00% | 11–40% | | Wide range of values: case study focus too broad |
| | | South Africa—citrus export growers | | 31.50% | <5% | | South Africa not typical of Sub-Saharan Africa |
| 4 | S&SE Asia—mango, guava, onions and other vegetables | India—fresh mango | very good | 17.20% | | 20.0% | |
| | | India—mango pulp | | | | 15.0% | |
| 5 | Industrialised Asia—roots and tubers | SW China—potatoes | poor | 34.70% | 5% (15–20% storage + transport) | | Inclusion of potatoes to feed animals as FW? |
| 6 | Latin America—roots and tubers | Latin America—cassava and potatoes | neither good nor poor | 26.40% | 7.0% | | Inclusion of potatoes to feed animals as FW? |
| 7 | Europe—oilseeds | France—oilseeds | good | 12.70% | | 6–10% | |
| 8 | Sub-Saharan Africa—groundnuts | Ethiopia and Malawi—groundnuts | very good | 11.70% | Manual production % loss, mechanised systems lower | | 17.8% |
| 9 | USA, Canada and Oceania—chickens | USA—broiler chickens, rearing and slaughter | poor | 6.20% | | 5.2–5.7% | |
| 10 | Sub-Saharan Africa—freshwater fisheries | S-SA Africa—Lake Victoria dagaa fishery | very poor | 49.00% | | 26–40% for PHL only | |

Half of the case-study supply chains found evidence from the literature or stakeholder interviews that supported the relevant loss rates used in the global estimates. Of the remainder, five cases provided research evidence that supported lower loss rates than those assumed in this study. For potatoes in industrialised Asia, potatoes and cassava in Latin America wheat in the UK, the differences may be partly explained by the inclusion of food surplus fed to animals within food waste definitions within the loss data used in global estimates. Many of the source studies focused on food availability and on-farm loss reduction rather than recording destinations relevant to the World Resources Institute and SDG 12.3 definition (Figure 3), which exclude the route to animal feed [12]. For citrus production in South Africa, divergence from the values used to represent Sub-Saharan Africa were explained by the citrus industry in that country having levels of food loss that were atypical of the region, with a highly developed sector driven by export markets and low loss rates achieved through high utilisation of fruit not conforming to export grade (for instance, within the juicing industry) [22].

For the Sub-Saharan groundnut case study, higher loss rates were reported within the case studies than those used in the study. However, the case-study areas in Malawi and Ethiopia focused on smallholder systems with highly manual cultivation techniques [23], whereas elsewhere in Sub-Saharan Africa, lower loss rates are apparent in more mechanised systems (e.g., within the major groundnut-producing countries that are major exporters, such as Nigeria and Sudan).

Although no adjustments to the global food-loss calculations were made based on the case-study findings, the analysis illustrates underlying uncertainties in relation to food-loss definitions and estimation of losses in primary production.

## 4. Discussion

The results from the re-evaluation of global food waste during primary production suggests that total losses are more extensive than previously estimated and that the pattern of losses across global regions is different to the findings of the 2011 FAO study. These

findings challenge the long-established assumption that farm-stage losses are a more significant issue within low-income regions and highlight some of the challenges in applying food waste definitions at the farm stage.

Data gaps and the lack of in-field measurement to underpin estimates remains a problem, although the analysis here has better data that available to the 2011 FAO study. Data meeting the selection criteria were unevenly spread across commodity groups and global regions, with cereals and fruit and vegetables better represented than others (particularly in Sub-Saharan Africa and South and Southeast Asia) and fish and dairy products having the fewest data points.

Apart from the lack of data points based on direct field measurements, a major uncertainty identified in half of the case studies related to inconsistent food-loss definitions. Formal food-loss definitions developed to clarify SDG 12.3 specify different 'destinations' of food losses to be excluded from food-loss definitions, such as feeding rejected produce to livestock. However, few loss studies at the agricultural stage have been conducted with these distinctions in mind, and this approach is less readily applied and less relevant to the working definitions of food loss and waste applied by those collecting data within the sector.

Within the limitations of the data, the study was able to differentiate harvest and post-harvest losses that occur at the farm stage. Overlap was identified between the two, and greater effort is required to include harvest losses within the scope of food-loss and waste reporting and tracking.

Diversion to animal feed was identified as a factor behind some of the discrepancies between food-loss calculations when sense checked against case-study findings. Although valorisation routes and diversion to animal feed reduce reported food waste, the objectives of improved food security and nutrition may be undermined in the process. Examples of non-food routes found in the case studies include the competition between dagaa fish supplied for human consumption and those used as fishmeal within livestock and aquaculture (case study 10), as well as oilseed for use in feed (case study 7). These routes often mask the full extent to which outputs from primary production are being underutilized as food and differ by commodity, farm system and level of mechanisation. While alternate markets can act as a sink for what might otherwise be counted as food loss, larger farms are also more likely to dedicate a higher proportion of their production to the supply to feed and processing uses, compared with smallholder farms (<2 ha), where growing crops for food predominates [24].

This is a complex area involving diversion of agricultural production to non-food uses as a planned farming activity (e.g., growing of wheat varieties intended for feed production), as well as an unintended consequence of poor growing conditions or last-minute order cancellations. Greater support for food markets over those in the feed sector is required to address this issue, but this is counteracted by the buoyant growth in diet transition towards dairy, fish and meat consumption in lower- and middle-income countries.

**Author Contributions:** Conceptualization, J.P. and T.C.; methodology, J.P. and T.C.; formal analysis, J.P. and T.C.; investigation and interviews, J.P. and A.B.; data curation, A.B.; writing—original draft preparation, J.P.; writing—review and editing, J.P. and A.B.; funding acquisition, J.P. All authors have read and agreed to the published version of the manuscript.

**Funding:** This research was commissioned by WWF-UK and funded through WWF-UK's partnership with Tesco.

**Institutional Review Board Statement:** Not applicable.

**Informed Consent Statement:** Not applicable.

**Data Availability Statement:** Data used in this study are available from the sources cited from within the article.

**Acknowledgments:** The authors would like to thank Lilly Da Gama, the WWF-UK and the Tesco project team, who provided support and direction to the project; Nicola Jenkin Pinpoint Sustainability for interviews conducted in South Africa; and Chloe McCloskey for project support.

**Conflicts of Interest:** The authors declare no conflict of interest.

## Appendix A

Classifications of food commodity groups used by the FAO Food Loss Index, FAO Food Balance Sheet and FAO 2011 food-loss study.

**Table A1.** Classifications of food commodity groups used by the FAO Food Loss Index, FAO Food Balance Sheet and FAO 2011 food-loss study [4].

| FAO 2019 Food Loss Index Commodity Groups | FAO Food Balance Sheet Groups | FAO 2011 Commodity Groups |
|---|---|---|
| Cereals and Pulses | Cereals | Cereals |
| | Pulses | Included within Oilseeds |
| Fruit and Vegetables | Fruit | Fruit and Vegetables |
| | Vegetables | |
| Roots, Tubers and Oil Crops | Oil Crops | Oilseeds and Pulses |
| | Roots and Tubers | Roots and Tubers |
| Meat and Animal Products | Animal fats | Meat and Poultry |
| | Eggs | |
| | Meat | |
| | Milk and Dairy | Dairy Products |
| Fish and Fish Products | Fish | Fish and Seafood |
| Other | Spices | Miscellaneous |
| | Stimulants | |
| | Sugars and Syrups | Sugars and Syrups |
| | Tree nuts | Included within Oilseeds and Pulses |

**Table A2.** Full listing of FAO commodities.

| Item | FAO Food Balance Sheet | FAO 2019 | FAO 2011 |
|---|---|---|---|
| Eggs, hen, in shell | Eggs | Meat and Animal Products | Meat and Poultry |
| Fat, camels | Animal Fats | Meat and Animal Products | Meat and Poultry |
| Fat, cattle | Animal Fats | Meat and Animal Products | Meat and Poultry |
| Fat, goats | Animal Fats | Meat and Animal Products | Meat and Poultry |
| Fat, sheep | Animal Fats | Meat and Animal Products | Meat and Poultry |
| Honey, natural | Sugar and Syrups | Other | Sugars and Syrups |
| Meat, camel | Meat | Meat and Animal Products | Meat and Poultry |
| Meat, cattle | Meat | Meat and Animal Products | Meat and Poultry |
| Meat, chicken | Meat | Meat and Animal Products | Meat and Poultry |
| Meat, game | Meat | Meat and Animal Products | Meat and Poultry |
| Meat, goat | Meat | Meat and Animal Products | Meat and Poultry |
| Meat, sheep | Meat | Meat and Animal Products | Meat and Poultry |

**Table A2.** *Cont.*

| Item | FAO Food Balance Sheet | FAO 2019 | FAO 2011 |
|---|---|---|---|
| Milk, whole fresh camel | Milk and Dairy | Meat and Animal Products | Dairy Products |
| Milk, whole fresh cow | Milk and Dairy | Meat and Animal Products | Dairy Products |
| Milk, whole fresh goat | Milk and Dairy | Meat and Animal Products | Dairy Products |
| Milk, whole fresh sheep | Milk and Dairy | Meat and Animal Products | Dairy Products |
| Offals, edible, camels | Meat | Meat and Animal Products | Meat and Poultry |
| Offals, edible, cattle | Meat | Meat and Animal Products | Meat and Poultry |
| Offals, edible, goats | Meat | Meat and Animal Products | Meat and Poultry |
| Offals, sheep, edible | Meat | Meat and Animal Products | Meat and Poultry |
| Eggs, other bird, in shell | Eggs | Meat and Animal Products | Meat and Poultry |
| Fat, pigs | Animal Fats | Meat and Animal Products | Meat and Poultry |
| Meatnot elsewhere specified | Meat | Meat and Animal Products | Meat and Poultry |
| Meat, pig | Meat | Meat and Animal Products | Meat and Poultry |
| Milk, whole fresh buffalo | Milk and Dairy | Meat and Animal Products | Dairy Products |
| Offals, pigs, edible | Meat | Meat and Animal Products | Meat and Poultry |
| Meat, horse | Meat | Meat and Animal Products | Meat and Poultry |
| Meat, rabbit | Meat | Meat and Animal Products | Meat and Poultry |
| Meat, turkey | Meat | Meat and Animal Products | Meat and Poultry |
| Meat, duck | Meat | Meat and Animal Products | Meat and Poultry |
| Meat, goose and guinea fowl | Meat | Meat and Animal Products | Meat and Poultry |
| Fat, buffaloes | Animal Fats | Meat and Animal Products | Meat and Poultry |
| Meat, buffalo | Meat | Meat and Animal Products | Meat and Poultry |
| Offals, edible, buffaloes | Meat | Meat and Animal Products | Meat and Poultry |
| Meat, other camelids | Meat | Meat and Animal Products | Meat and Poultry |
| Meat, other rodents | Meat | Meat and Animal Products | Meat and Poultry |
| Meat, ass | Meat | Meat and Animal Products | Meat and Poultry |
| Meat, bird not elsewhere specified | Meat | Meat and Animal Products | Meat and Poultry |
| Meat, mule | Meat | Meat and Animal Products | Meat and Poultry |
| Snails, not sea | Meat | Meat and Animal Products | Meat and Poultry |
| Almonds, with shell | Tree Nuts | Other | Oilseeds and Pulses |
| Anise, badian, fennel, coriander | Spices | Other | Miscellaneous |
| Apples | Fruit | Fruit and Vegetables | Fruit and Vegetables |
| Apricots | Fruit | Fruit and Vegetables | Fruit and Vegetables |
| Barley | Cereals | Cereals and Pulses | Cereals |
| Berries not elsewhere specified | Fruit | Fruit and Vegetables | Fruit and Vegetables |
| Cottonseed | Oil Crops | Roots, Tubers and Oil Crops | Oilseeds and Pulses |

**Table A2.** *Cont.*

| Item | FAO Food Balance Sheet | FAO 2019 | FAO 2011 |
|---|---|---|---|
| Figs | Fruit | Fruit and Vegetables | Fruit and Vegetables |
| Fruit, citrus not elsewhere specified | Fruit | Fruit and Vegetables | Fruit and Vegetables |
| Fruit, fresh not elsewhere specified | Fruit | Fruit and Vegetables | Fruit and Vegetables |
| Fruit, stone not elsewhere specified | Fruit | Fruit and Vegetables | Fruit and Vegetables |
| Grapes | Fruit | Fruit and Vegetables | Fruit and Vegetables |
| Linseed | Oil Crops | Roots, Tubers and Oil Crops | Oilseeds and Pulses |
| Maize | Cereals | Cereals and Pulses | Cereals |
| Melons, other (inc. cantaloupes) | Fruit | Fruit and Vegetables | Fruit and Vegetables |
| Millet | Cereals | Cereals and Pulses | Cereals |
| Nuts not elsewhere specified | Tree Nuts | Other | Oilseeds and Pulses |
| Olives | Oil Crops | Roots, Tubers and Oil Crops | Oilseeds and Pulses |
| Onions, dry | Vegetables | Fruit and Vegetables | Fruit and Vegetables |
| Oranges | Fruit | Fruit and Vegetables | Fruit and Vegetables |
| Millet | Cereals | Cereals and Pulses | Cereals |
| Nuts not elsewhere specified | Tree Nuts | Other | Oilseeds and Pulses |
| Olives | Oil Crops | Roots, Tubers and Oil Crops | Oilseeds and Pulses |
| Onions, dry | Vegetables | Fruit and Vegetables | Fruit and Vegetables |
| Oranges | Fruit | Fruit and Vegetables | Fruit and Vegetables |
| Peaches and nectarines | Fruit | Fruit and Vegetables | Fruit and Vegetables |
| Pears | Fruit | Fruit and Vegetables | Fruit and Vegetables |
| Pistachios | Tree Nuts | Other | Oilseeds and Pulses |
| Plums and sloes | Fruit | Fruit and Vegetables | Fruit and Vegetables |
| Potatoes | Roots and Tubers | Roots, Tubers and Oil Crops | Roots and Tubers |
| Pulses not elsewhere specified | Pulses | Cereals and Pulses | Oilseeds and Pulses |
| Rice, paddy | Cereals | Cereals and Pulses | Cereals |
| Sesame seed | Oil Crops | Roots, Tubers and Oil Crops | Oilseeds and Pulses |
| Spicesnot elsewhere specified | Spices | Other | Miscellaneous |
| Sugar beet | Sugar and Syrups | Other | Sugars and Syrups |
| Sugar cane | Sugar and Syrups | Other | Sugars and Syrups |
| Sunflower seed | Oil Crops | Roots, Tubers and Oil Crops | Oilseeds and Pulses |
| Vegetables, freshnot elsewhere specified | Vegetables | Fruit and Vegetables | Fruit and Vegetables |
| Walnuts, with shell | Tree Nuts | Other | Oilseeds and Pulses |
| Watermelons | Fruit | Fruit and Vegetables | Fruit and Vegetables |
| Wheat | Cereals | Cereals and Pulses | Cereals |

**Table A2.** *Cont.*

| Item | FAO Food Balance Sheet | FAO 2019 | FAO 2011 |
|---|---|---|---|
| Beans, dry | Pulses | Cereals and Pulses | Oilseeds and Pulses |
| Beans, green | Vegetables | Fruit and Vegetables | Fruit and Vegetables |
| Broad beans, horse beans, dry | Pulses | Cereals and Pulses | Oilseeds and Pulses |
| Cabbages and other brassicas | Vegetables | Fruit and Vegetables | Fruit and Vegetables |
| Carrots and turnips | Vegetables | Fruit and Vegetables | Fruit and Vegetables |
| Cauliflowers and broccoli | Vegetables | Fruit and Vegetables | Fruit and Vegetables |
| Cherries | Fruit | Fruit and Vegetables | Fruit and Vegetables |
| Cherries, sour | Fruit | Fruit and Vegetables | Fruit and Vegetables |
| Chestnut | Tree Nuts | Other | Oilseeds and Pulses |
| Chillies and peppers, green | Vegetables | Fruit and Vegetables | Fruit and Vegetables |
| Cucumbers and gherkins | Vegetables | Fruit and Vegetables | Fruit and Vegetables |
| Dates | Fruit | Fruit and Vegetables | Fruit and Vegetables |
| Eggplants (aubergines) | Vegetables | Fruit and Vegetables | Fruit and Vegetables |
| Garlic | Vegetables | Fruit and Vegetables | Fruit and Vegetables |
| Leeks, other alliaceous vegetables | Vegetables | Fruit and Vegetables | Fruit and Vegetables |
| Lemons and limes | Fruit | Fruit and Vegetables | Fruit and Vegetables |
| Lettuce and chicory | Vegetables | Fruit and Vegetables | Fruit and Vegetables |
| Mushrooms and truffles | Vegetables | Fruit and Vegetables | Fruit and Vegetables |
| Oats | Cereals | Cereals and Pulses | Cereals |
| Okra | Vegetables | Fruit and Vegetables | Fruit and Vegetables |
| Onions, shallots, green | Vegetables | Fruit and Vegetables | Fruit and Vegetables |
| Peas, green | Vegetables | Fruit and Vegetables | Fruit and Vegetables |
| Pumpkins, squash and gourds | Vegetables | Fruit and Vegetables | Fruit and Vegetables |
| Quinces | Fruit | Fruit and Vegetables | Fruit and Vegetables |
| Rye | Cereals | Cereals and Pulses | Cereals |
| Sorghum | Cereals | Cereals and Pulses | Cereals |
| Soybeans | Oil Crops | Roots, Tubers and Oil Crops | Oilseeds and Pulses |
| Spinach | Vegetables | Fruit and Vegetables | Fruit and Vegetables |
| Strawberries | Fruit | Fruit and Vegetables | Fruit and Vegetables |
| Tangerines, mandarins, clementines, satsumas | Fruit | Fruit and Vegetables | Fruit and Vegetables |
| Tomatoes | Vegetables | Fruit and Vegetables | Fruit and Vegetables |
| Vegetables, leguminousnot elsewhere specified | Vegetables | Fruit and Vegetables | Fruit and Vegetables |
| Vetches | Pulses | Cereals and Pulses | Oilseeds and Pulses |
| Artichokes | Vegetables | Fruit and Vegetables | Fruit and Vegetables |
| Bananas | Fruit | Fruit and Vegetables | Fruit and Vegetables |
| Carobs | Vegetables | Fruit and Vegetables | Fruit and Vegetables |
| Chickpeas | Pulses | Cereals and Pulses | Oilseeds and Pulses |

**Table A2.** *Cont.*

| Item | FAO Food Balance Sheet | FAO 2019 | FAO 2011 |
|---|---|---|---|
| Chilis and peppers, dry | Spices | Other | Miscellaneous |
| Fruit, tropical freshnot elsewhere specified | Fruit | Fruit and Vegetables | Fruit and Vegetables |
| Grapefruit (inc. pomelos) | Fruit | Fruit and Vegetables | Fruit and Vegetables |
| Groundnuts, with shell | Oil Crops | Roots, Tubers and Oil Crops | Oilseeds and Pulses |
| Lentils | Pulses | Cereals and Pulses | Oilseeds and Pulses |
| Peas, dry | Pulses | Cereals and Pulses | Oilseeds and Pulses |
| Rapeseed | Oil Crops | Roots, Tubers and Oil Crops | Oilseeds and Pulses |
| Triticale | Cereals | Cereals and Pulses | Cereals |
| Cassava | Roots and Tubers | Roots, Tubers and Oil Crops | Roots and Tubers |
| Cocoa, beans | Stimulants | Other | Miscellaneous |
| Coconuts | Oil Crops | Roots, Tubers and Oil Crops | Oilseeds and Pulses |
| Maize, green | Vegetables | Fruit and Vegetables | Fruit and Vegetables |
| Pineapples | Fruit | Fruit and Vegetables | Fruit and Vegetables |
| Taro (cocoyam) | Roots and tubers | Roots, Tubers and Oil Crops | Roots and Tubers |
| Yams | Roots and tubers | Roots, Tubers and Oil Crops | Roots and Tubers |
| Cashew nuts, with shell | Tree Nuts | Other | Oilseeds and Pulses |
| Castor oil seed | Oil Crops | Roots, Tubers and Oil Crops | Oilseeds and Pulses |
| Coffee, green | Stimulants | Other | Miscellaneous |
| Oil palm fruit | Oil Crops | Roots, Tubers and Oil Crops | Oilseeds and Pulses |
| Oil, palm | Oil Crops | Roots, Tubers and Oil Crops | Oilseeds and Pulses |
| Palm kernels | Oil Crops | Roots, Tubers and Oil Crops | Oilseeds and Pulses |
| Sweet potatoes | Roots and Tubers | Roots, Tubers and Oil Crops | Roots and Tubers |
| Mangoes, mangosteens, guavas | Fruit | Fruit and Vegetables | Fruit and Vegetables |
| Asparagus | Vegetables | Fruit and Vegetables | Fruit and Vegetables |
| Avocados | Fruit | Fruit and Vegetables | Fruit and Vegetables |
| Canary seed | Cereals | Cereals and Pulses | Cereals |
| Cerealsnot elsewhere specified | Cereals | Cereals and Pulses | Cereals |
| Lupins | Pulses | Cereals and Pulses | Oilseeds and Pulses |
| Maté | Stimulants | Other | Miscellaneous |
| Oilseedsnot elsewhere specified | Oil Crops | Roots, Tubers and Oil Crops | Oilseeds and Pulses |
| Papayas | Fruit | Fruit and Vegetables | Fruit and Vegetables |
| Peppermint | Spices | Other | Miscellaneous |
| Safflower seed | Oil Crops | Roots, Tubers and Oil Crops | Oilseeds and Pulses |
| String beans | Vegetables | Fruit and Vegetables | Fruit and Vegetables |
| Tea | Stimulants | Other | Miscellaneous |
| Tung nuts | Oil Crops | Roots, Tubers and Oil Crops | Oilseeds and Pulses |
| Hazelnuts, with shell | Tree Nuts | Other | Oilseeds and Pulses |

**Table A2.** *Cont.*

| Item | FAO Food Balance Sheet | FAO 2019 | FAO 2011 |
|---|---|---|---|
| Blueberries | Fruit | Fruit and Vegetables | Fruit and Vegetables |
| Cow peas, dry | Pulses | Cereals and Pulses | Oilseeds and Pulses |
| Currants | Fruit | Fruit and Vegetables | Fruit and Vegetables |
| Kiwi fruit | Fruit | Fruit and Vegetables | Fruit and Vegetables |
| Mustard seed | Oil Crops | Roots, Tubers and Oil Crops | Oilseeds and Pulses |
| Persimmons | Fruit | Fruit and Vegetables | Fruit and Vegetables |
| Raspberries | Fruit | Fruit and Vegetables | Fruit and Vegetables |
| Buckwheat | Cereals | Cereals and Pulses | Cereals |
| Gooseberries | Fruit | Fruit and Vegetables | Fruit and Vegetables |
| Grain, mixed | Cereals | Cereals and Pulses | Cereals |
| Poppy seed | Oil Crops | Roots, Tubers and Oil Crops | Oilseeds and Pulses |
| Cranberries | Fruit | Fruit and Vegetables | Fruit and Vegetables |
| Pigeon peas | Pulses | Cereals and Pulses | Oilseeds and Pulses |
| Plantains and others | Fruit | Fruit and Vegetables | Fruit and Vegetables |
| Areca nuts | Tree Nuts | Other | Oilseeds and Pulses |
| Ginger | Spices | Other | Miscellaneous |
| Sugar cropsnot elsewhere specified | Sugar and Syrups | Other | sugars and syrups |
| Fruit, pomenot elsewhere specified | Fruit | Fruit and Vegetables | Fruit and Vegetables |
| Chicory roots | Vegetables | Fruit and Vegetables | Fruit and Vegetables |
| Vanilla | Spices | Other | Miscellaneous |
| Roots and tubersnot elsewhere specified | Roots and Tubers | Roots, Tubers and Oil Crops | Roots and Tubers |
| Yautia (cocoyam) | Roots and Tubers | Roots, Tubers and Oil Crops | Roots and Tubers |
| Fonio | Cereals | Cereals and Pulses | Cereals |
| Karite nuts (shea nuts) | Oil Crops | Roots, Tubers and Oil Crops | Oilseeds and Pulses |
| Kola nuts | Tree Nuts | Other | Oilseeds and Pulses |
| Melon seed | Oil Crops | Roots, Tubers and Oil Crops | Oilseeds and Pulses |
| Pepper (piper spp.) | Spices | Other | Miscellaneous |
| Nutmeg, mace and cardamoms | Spices | Other | Miscellaneous |
| Brazil nuts, with shell | Tree Nuts | Other | Oilseeds and Pulses |
| Pyrethrum, dried | Spices | Other | Miscellaneous |
| Quinoa | Cereals | Cereals and Pulses | Cereals |
| Cashew apple | Fruit | Fruit and Vegetables | Fruit and Vegetables |
| Cassava leaves | Vegetables | Fruit and Vegetables | Fruit and Vegetables |
| Hempseed | Oil Crops | Roots, Tubers and Oil Crops | Oilseeds and Pulses |
| Bambara beans | Pulses | Cereals and Pulses | Oilseeds and Pulses |
| Cinnamon (cannella) | Spices | Other | Miscellaneous |

**Table A2.** *Cont.*

| Item | FAO Food Balance Sheet | FAO 2019 | FAO 2011 |
|---|---|---|---|
| Cloves | Spices | Other | Miscellaneous |
| Tallow tree seed | Oil Crops | Roots, Tubers and Oil Crops | Oilseeds and Pulses |
| Kapok fruit | Oil Crops | Roots, Tubers and Oil Crops | Oilseeds and Pulses |
| Kapok seed in shell | Oil Crops | Roots, Tubers and Oil Crops | Oilseeds and Pulses |
| Jojoba seed | Oil Crops | Roots, Tubers and Oil Crops | Oilseeds and Pulses |

## Appendix B

Geographical groupings used in food-loss analysis.

**Table A3.** Countries included in high- and medium–income 'industrialised' regions.

| Europe | | |
|---|---|---|
| Albania | Georgia | Netherlands |
| Armenia | Germany | Norway |
| Austria | Greece | Poland |
| Azerbaijan | Hungary | Portugal |
| Belarus | Iceland | Romania |
| Belgium | Ireland | Russian Federation |
| Bosnia & Herzegovina | Italy | Serbia |
| Bulgaria | Latvia | Slovakia |
| Croatia | Lithuania | Slovenia |
| Cyprus | Luxemburg | Spain |
| Czech Republic | Macedonia | Sweden |
| Denmark | Malta | Switzerland |
| Estonia | Moldova | Ukraine |
| Finland | Montenegro | United Kingdom |
| France | | |
| North America and Oceania (NA&Oce) | | Industrialized Asia (Ind. Asia) |
| Australia | USA | Japan |
| Canada | | China |
| New Zealand | | Republic of Korea |

**Table A4.** Countries included in low-income regions.

| Sub-Saharan Africa | | North Africa, Western and Central Asia | South and Southeast Asia | Latin America |
|---|---|---|---|---|
| Angola | Malawi | Algeria | Afghanistan | Argentina |
| Benin | Mali | Egypt | Bangladesh | Belize |
| Botswana | Mauritania | Iraq | Bhutan | Bolivia |
| Burkina Faso | Mozambique | Israel | Cambodia | Brazil |
| Burundi | Namibia | Jordan | India | Chile |

**Table A4.** *Cont.*

| Sub-Saharan Africa | | North Africa, Western and Central Asia | South and Southeast Asia | Latin America |
|---|---|---|---|---|
| Cameroon | Niger | Kazakhstan | Indonesia | Colombia |
| Central African Rep | Nigeria | Kuwait | Iran | Costa Rica |
| Chad | Rwanda | Kyrgyzstan | Laos | Cuba |
| Dem Rep of Congo | Senegal | Lebanon | Malaysia | Dominican Republic |
| Cote d'Ivoire | Sierra Leone | Libya | Myanmar | Ecuador |
| Equatorial Guinea | Somalia | Mongolia | Nepal | El Salvador |
| Eritrea | South Africa | Morocco | Pakistan | Guatemala |
| Ethiopia | Sudan | Oman | Philippines | Guyana |
| Gabon | Swaziland | Saudi Arabia | Sri Lanka | Haiti |
| Gambia | Tanzania | Syria | Thailand | Honduras |
| Ghana | Togo | Tajikistan | Vietnam | Jamaica |
| Guinea | Uganda | Tunisia | | Mexico |
| Guinea-Bissau | Zambia | Turkey | | Nicaragua |
| Kenya | Zimbabwe | Turkmenistan | | Panama |
| Lesotho | | United Arab Emirates | | Paraguay |
| Liberia | | Uzbekistan | | Peru |
| | | Yemen | | Suriname |
| | | | | Uruguay |
| | | | | Venezuela |

## Appendix C

**Table A5.** Commodity-region case studies.

| | Commodity/Region | Evidence Collected: Interviews Conducted for the Research and Literature Reviews |
|---|---|---|
| Cereals and pulses | 1. European—wheat production in UK | Interview with trade association, literature review—10 references. |
| | 2. S&SE Asia—rice production | Interview with in-country experts with 14–15 years' experience working on rice crops (WWF team) and use of literature focusing on losses in India and Pakistan—5 references. |
| Fruit and vegetables | 3. Sub-Saharan Africa—citrus fruit, tomato and other vegetables | 5 interviews covering different components of citrus production—growers, trade bodies, exporters and academic research sector. Literature review as primary source exploring losses for smallholder farms—9 references. |
| | 4. S&SE Asia—mango, guava, aubergine, onions and other vegetables | Interview and literature review—mango in India and detailed mapping within Andhra Pradesh—7 references. |
| Roots, tubers and oil crops | 5. Industrialised Asia—potato and sweet potato in SW China | Interview—researcher with potato tuber expertise in industrialised Asia working with farmers—4 references. |
| | 6. Latin America—cassava, potato and sweet potato production | Interview relating to losses within Peru, literature with a focus on Trinidad and Tobago, Guyana (cassava) and Peru (potato)—3 references. |
| | 7. Europe—rape seed and sunflower seed | France: oilseeds—4 references. |
| | 8. Sub-Saharan Africa—groundnuts | Interview with researcher and groundnut co-ordinator for Ethiopia, additional literature from Malawi—5 references. |
| Meat and animal products | 9. USA, Canada and Oceania—broiler-chicken rearing/slaughter | Interview with meat sector expert/consultant: USA, broiler chickens—7 references |

**Table A5.** *Cont.*

| | Commodity/Region | Evidence Collected: Interviews Conducted for the Research and Literature Reviews |
|---|---|---|
| Fish and seafood | 10. Sub-Saharan Africa—freshwater fisheries | East Africa, Lake Victoria dagaa fishery—11 references. |
| Overarching issues | A series of interviews conducted to explore overarching issues in relation to farm-stage losses | 7 interviews, including an NGO working on farm-stage losses associated with crops exported to UK from Africa and Latin America; 2 interviews with conservation charity-policy officer working on food loss, academic expert on farm-stage food-loss measurement, retailer working on Champions 12.3 10∗20∗30 initiative, researchers developing food-loss solutions for fruit and vegetables, researcher within government department responsible for food-loss reporting. |

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
