# Peer review of "Global Food Loss and Waste in Primary Production: A Reassessment of Its Scale and Significance"

_sustainability, doi:10.3390/su132112087_

Round 1

Reviewer 1 Report

I still can’t understand how the authors obtained the data they present. This can happen because there is indeed a lot of missing parts in the paper.

 When you compare this paper with one they cite, and oppose, namely

Gustavsson, J. et al. The Methodology of the FAO study: Global food losses and food waste – extent, causes and prevention – FAO, 2011

you can see that, in the second, the models, assumptions, conversion factors used to relate the FAO production data to the food waste data from literature are described. This does not happen in the first.

To cite some examples:

  • On lines 110 -111, the authors stated that the inedible components from food were excluded from food waste totals. How did they do that? Considering that most of the data came from interviews (Figure 2), they should have used some conversion factor since when farms and experts think of banana waste, they think of it as pulp + skin waste.
  • On lines 131 to 135, the authors informed they used estimates of waste for specified commodity and geographies rather than generic groupings such as fruit and vegetables. In such a diverse group as fruit and vegetables, for example, with much less data on food waste when you compare with grains, how did the authors go from individual fruits and vegetables data to the result they present on Figure 5 ?
  • On lines 165-166, Table 2. If I understood it right, the authors inform that the number of studies meeting selection criteria for pulses and cereals were 2 in North America & Australasia, 23 in Latin America and more than 1.000 in Sub-Saharan Africa. I could not understand how one can have more data for cereal and pulse waste at farm level in Sub-Saharan Africa compared to data in countries like USA, Canada, Brazil and Argentina, that feed the world with the grains they produce in very organized production chains, more likely to have sound data on this food category waste.  
  • The triangulation case studies were selected based on relevance and representativeness or were the ones available in literature? Reading lines 214-217, I understood the authors had conducted the case studies, but later on Appendix C, there are a number of references mentioned in the column ‘Evidence collected’, what suggested that they did not do it themselves.
  • I also could not understand how the triangulation case studies improved the data collected otherwise.

Author Response

  • On lines 110 -111, the authors stated that the inedible components from food were excluded from food waste totals. How did they do that? Considering that most of the data came from interviews (Figure 2), they should have used some conversion factor since when farms and experts think of banana waste, they think of it as pulp + skin waste.
  • RESPONSE: Further clarifications have been added to the text. The quantitative data were not collected through interviews and the allocation factors were derived from the FAO 2011 study, as were the conversion factors used to estimate the edible fractions.
  • On lines 131 to 135, the authors informed they used estimates of waste for specified commodity and geographies rather than generic groupings such as fruit and vegetables. In such a diverse group as fruit and vegetables, for example, with much less data on food waste when you compare with grains, how did the authors go from individual fruits and vegetables data to the result they present on Figure 5 ?

    RESPONSE: Where there were data gaps or insufficient representation for a commodity group, appropriate substitute values were used, based on data obtained from comparable regions or from the closest commodity group. For a diverse group, such as fruit and vegetables, the study used proxy values derived from similar produce in terms of perishability, type and agricultural system. 

  • On lines 165-166, Table 2. The authors inform that the number of studies meeting selection criteria for pulses and cereals were 2 in North America & Australasia, 23 in Latin America and more than 1.000 in Sub-Saharan Africa. I could not understand how one can have more data for cereal and pulse waste at farm level in Sub-Saharan Africa compared to data in countries like USA, Canada, Brazil and Argentina, that feed the world with the grains they produce in very organized production chains, more likely to have sound data on this food category waste. 
  • RESPONSE: This is correct: many food loss studies have been conducted in global regions where food access/ security is a key concern. These regions have generated far more field loss data than is available for major grain producing regions of the world. Highly industrialised systems look at overall production yield but do not directly measure grain losses in the field as this information is too time-consuming to collect. The 2011 FAO study was only able to find 1 data point relating to the field losses associated with North American wheat production.
  • The triangulation case studies were selected based on relevance and representativeness or were the ones available in literature? Reading lines 214-217, I understood the authors had conducted the case studies, but later on Appendix C, there are a number of references mentioned in the column ‘Evidence collected’, what suggested that they did not do it themselves.
  • RESPONSE: The study carried out 20 interviews across different stakeholder groups, including with NGOs, trade associations, primary producers, and research institutes. Of these 13 interviews were specific to commodity-regions and 7 explored over-arching themes, such as field measurement, whole chain initiatives and food waste drivers of farm stage losses. As expertise relating to farm stage losses is fragmented and not easily accessed, so it was not possible to complete interviews for all ten case studies. Further evidence gathering involved an extensive literature review that the research team carried out for each of the 10 commodity-regions. A clarification has been added to the table in Appendix C making it clear that this element involved primary data collection carried out by the research team.
  • I also could not understand how the triangulation case studies improved the data collected otherwise.
  • RESPONSE: The purpose of the case studies was as a sense-check on the food loss calculations and to look at the definitional issues, as summarised in Table 5. 

Reviewer 2 Report

Dear authors,
the contribution addresses an extremely interesting topic that aims to analyze and describe global food losses and waste in primary production: a reassessment of its scope and significance.From my point of view, I would recommend writing more about how research can bridge the void in literature. I would add the chapter on literature review. In the introduction, you could deepen the topic you are dealing with a little more generally, also with reference to previous studies. Please discuss the literature review in a pro-versus approach, highlighting the benefits of different approaches in the field while discussing the limitations of the research. I would go deeper into the research questions. I would develop the implications for theory, practice and policymakers in one chapter. I recommend adding sources to figures and tables in the text. Increase references. I appreciate the effort of the authors and hope they will find these comments helpful in improving their research article in the future.

Best regards

Author Response

I would recommend writing more about how research can bridge the void in literature. I would add the chapter on literature review. In the introduction, you could deepen the topic you are dealing with a little more generally, also with reference to previous studies.

RESPONSE: Options to address the paucity of global food loss data was not an aspect that the team researched in any detail, as the focus of the study was on the reassessment of global farm stage losses, making the best use of available data.  Further research is planned on the more qualitative aspects looking at the linkages between food losses within primary production and factors driving losses at farm stage and from within the supply chain and end markets.

Please discuss the literature review in a pro-versus approach, highlighting the benefits of different approaches in the field while discussing the limitations of the research. I would go deeper into the research questions. I would develop the implications for theory, practice and policymakers in one chapter. I recommend adding sources to figures and tables in the text. Increase references.

RESPONSE: The study that we were commissioned to undertake did not review the different approaches to measurement in any detail, nor did we conduct an extensive literature review into more theoretical aspects.  The majority of the research effort was expended on the collation and shifting of 20,000 data points and the development of a framework calculating losses.

Reviewer 3 Report

Chosen theme of the article is actual. Authors used very rich material consisted of statistical data. Used literature is suitable. The article covered analyzed topic in full extense. The article is written on good level.

     Partial goal of the article was to identify the scale and global profile of farm stage food losses.  Greater support to food markets over those in the feed sector is required, but this is counter-acted by the buoyant growth in diet transition towards dairy, fish and meat consumption in lower and middle income countries.

     Goals of the article given by authors were fulfilled. Authors´s points of view are suitably expressed. Applied metodology was previously based on comparative and descriptive methods.

Author Response

Thank you very much for your review.

Round 2

Reviewer 2 Report

Dear Authors,
it was adequately and thoroughly explained how the research was conducted.
The manuscript can be accepted in its current form.
Best regards